# Exploring protein hotspots by optimized fragment pharmacophores

Dávid Bajusz [1], Warren S. Wade[2], Grzegorz Satała[3], Andrzej J. Bojarski[3], Janez Ilaš [4], Jessica Ebner[5], Florian Grebien [5], Henrietta Papp [6], Ferenc Jakab[6], Alice Douangamath [7,8], Daren Fearon [7,8], Frank von Delft [7,8,9,10,11], Marion Schuller[12], Ivan Ahel[12], Amanda Wakefield[13,14], Sándor Vajda[13,14], János Gerencsér [2], Péter Pallai[2] & György M. Keserű [1✉]

Fragment-based drug design has introduced a bottom-up process for drug development, with improved sampling of chemical space and increased effectiveness in early drug discovery. Here, we combine the use of pharmacophores, the most general concept of representing drug-target interactions with the theory of protein hotspots, to develop a design protocol for fragment libraries. The SpotXplorer approach compiles small fragment libraries that maximize the coverage of experimentally confirmed binding pharmacophores at the most preferred hotspots. The efficiency of this approach is demonstrated with a pilot library of 96 fragment-sized compounds (SpotXplorer0) that is validated on popular target classes and emerging drug targets. Biochemical screening against a set of GPCRs and proteases retrieves compounds containing an average of 70% of known pharmacophores for these targets. More importantly, SpotXplorer0 screening identifies confirmed hits against recently established challenging targets such as the histone methyltransferase SETD2, the main protease (3CLPro) and the NSP3 macrodomain of SARS-CoV-2.

[1] Medicinal Chemistry Research Group, Research Centre for Natural Sciences, Budapest, Hungary. [2] BioBlocks, Inc., San Diego, CA, USA. [3] Maj Institute of Pharmacology Polish Academy of Sciences, Kraków, Poland. [4] Faculty of Pharmacy, University of Ljubljana, Ljubljana, Slovenia. [5] Institute for Medical Biochemistry, University of Veterinary Medicine, Vienna, Austria. [6] National Laboratory of Virology, Szentágothai Research Centre, University of Pécs, Pécs, Hungary. [7] Diamond Light Source Ltd., Harwell Science and Innovation Campus, Didcot, UK. [8] Research Complex at Harwell, Harwell Science and Innovation Campus, Didcot OX11 0FA, UK. [9] Structural Genomics Consortium, University of Oxford, Old Road Campus, Roosevelt Drive, Headington OX3 7DQ, UK. [10] Centre for Medicines Discovery, University of Oxford, Old Road Campus, Roosevelt Drive, Headington OX3 7DQ, UK. [11] Department of Biochemistry, University of Johannesburg, Auckland Park 2006, South Africa. [12] Sir William Dunn School of Pathology, University of Oxford, Oxford, UK. [13] Department of Chemistry, Boston University, Boston, MA, USA. [14] Department of Biomedical Engineering, Boston University, Boston, MA, USA. ✉email: keseru.gyorgy@ttk.hu

Fragment-based drug discovery (FBDD) is a lead generation strategy based on the screening of small polar compounds that typically exhibit low affinity towards protein targets. Previous binding thermodynamics studies revealed that fragments bind to protein hotspots by choice[1–3]. Their polar character prevents apolar desolvation compensation for rigid body entropy loss[4] and generates productive fragment binding. Indeed, fragments have been shown to bind to protein hotspots by a limited number of optimal geometry H-bonds[5,6]. Good overlap with the structure-based pharmacophores at the primary hotspot (binding site region with the largest contribution to the ligand binding free energy) leads to robust binding[3]. Therefore the nature and quality of binding interactions is fundamental to the choice of fragment starting points for drug discovery programs[7]. Recent studies have shown that there is still significant controversy on the applicability of the fragment-based approach to drug discovery[3,8]. When reviewing fragment starting points, target specificity was often identified as an issue. An analysis of the active fragments from 35 campaigns on 20 targets at Novartis indicated that 63% of the screened fragments had never been observed as hits[9]. Instead, the team promoted privileged fragments as high-value library members active on more than one target. Another analysis found that 20% of fragments in crystallized complexes in the Protein Data Bank (PDB) have multiple protein targets[10]. Since there is often an opportunity to build selectivity while evolving fragments[11], features encoded in frequent hitter fragments can productively be considered part of a representative set of binding pharmacophores. The evolutionarily conserved nature of binding hotspots[12] and the conservation of pharmacophores found in certain target classes[13], suggests that the number of distinct structure-based pharmacophores should be limited. Our idea was to identify the set of fragment pharmacophores that covers the diversity of known hotspots and use them to design a small, efficient fragment library to recognize this diversity.

In the present work, we analyze critical interactions found at the hotspot of target proteins to derive fragment pharmacophores represented in fragment-protein complexes available in the PDB. Using this information, we design a minimal diverse set of commercial fragments (the SpotXplorer0 library) covering the majority of the experimental binding pharmacophores to be used for the identification of fragment starting points for drug discovery targets. After validating this library against established target classes and reproducing a majority of the known binding pharmacophores, we address more challenging targets. Besides covering a high percentage of known pharmacophores of well-established drug targets, our approach identified the first non-nucleoside type inhibitor for the histone methyltransferase SET domain containing 2 (SETD2), an emerging leukemia target[14]. Furthermore, we present promising starting points for two recent COVID-19 targets, the 3CL main protease and the NSP3 macrodomain of SARS-CoV-2.

## Results

### Experimental fragment-binding modes are represented by a limited set of pharmacophores.
Experimental fragment pharmacophores were extracted from the available protein-fragment structures in the Protein Data Bank. More than 3300 PDB entries with fragment-sized ligands, containing 10–16 heavy atoms[15], were filtered and analyzed in a stepwise workflow reported in Fig. 1 and, in more detail, in the Supplementary Information, section 1. The FTMap approach (ATLAS software)[16] was utilized to identify fragment-sized ligands in a binding hotspot of the target protein[16,17]. FTMap is an established protein mapping algorithm that predicts the location and strength of hotspots

within a protein. The method distributes small organic probe molecules of varying size, shape, and polarity on a dense grid defined on the protein surface, finds the most favorable positions for each probe type, and performs local energy minimization allowing for probe flexibility. Then, it clusters the probes and ranks the clusters on the basis of their average energy. Regions that bind multiple probe clusters are defined as the predicted binding hotspots. (The method is described in more detail in the Supplementary Information, section 1.3.) Schrödinger's ePharmacophore module was then employed to extract a pharmacophore model for each protein–ligand complex. A maximum of four available features having the largest energetic contributions to protein–ligand binding[18,19] were extracted for each ligand.

In many cases, more than one protein-fragment complex was available for a single protein target, often with similar binding modes and pharmacophores. Moreover, binding pharmacophores were often conserved even at the fragment level among groups of similar or related proteins, with the kinase hinge binding motif being one of the most well-known examples[20]. Consequently, many of the 3584 extracted pharmacophore models were redundant or very similar, supporting our expectation that the number of distinct binding pharmacophores was relatively small. Therefore, we applied a two-step clustering process to identify a non-redundant set of pharmacophores. First, pharmacophores containing the exact same set of features were grouped together into 141 level 1 clusters, e.g., all pharmacophores that contain an H-bond donor (D) and two aromatic ring (R) features belong to the DRR group (one letter for each feature in alphabetical order). Second, members of each such group were spatially aligned to each other and a pairwise root-mean-squared distance (RMSD) matrix was calculated and processed by a hierarchical clustering (HCA) algorithm with the complete linkage rule and an RMSD cutoff of 2 Å. The resulting clusters were labeled with sequential numbers (e.g., DRR_0, DRR_1, etc.) that denote a specific 3D arrangement of the features. In total, hotspot-binding fragments of the sampled fragment-protein complexes contain 425 non-redundant binding pharmacophores (the top 20 most populated pharmacophores are included in Supplementary Fig. 2). Even a small fragment library featuring all 425 pharmacophores could thus enable fragment hit discovery against a wide range of protein targets.

### Optimizing a small fragment library for pharmacophore coverage and diversity.
A first generation fragment library was designed to cover most of the base set of binding pharmacophores identified from protein-fragment X-ray structures. By definition, fragments have small sizes and a limited number of pharmacophore features[21] that form only a few characteristic interactions with their targets[5]. Therefore, we focused our optimization on the identification of a minimal set of fragments covering the largest possible fraction of experimentally validated 2-point and 3-point binding pharmacophores. Fragment collections from BioBlocks and other commercially available sources were filtered for size, rotatable bond count, and other properties (Supplementary Information, section 2.1), and the absence of known problematic features[22], and matched to the 425 non-redundant pharmacophore models. An important post-processing step was the detection of submodels. Essentially, if a smaller pharmacophore is present as a spatially matching subset of a larger pharmacophore (Fig. 1b), then the molecule is considered as a match for only the larger one. This way, smaller pharmacophores are represented on their own right, rather than trivially being present in molecules with a matching larger pharmacophore (see Fig. 1c, and for more detail, see section 1.5 of the Supplementary Information). We stored the results in a 425-bit pharmacophore fingerprint in two

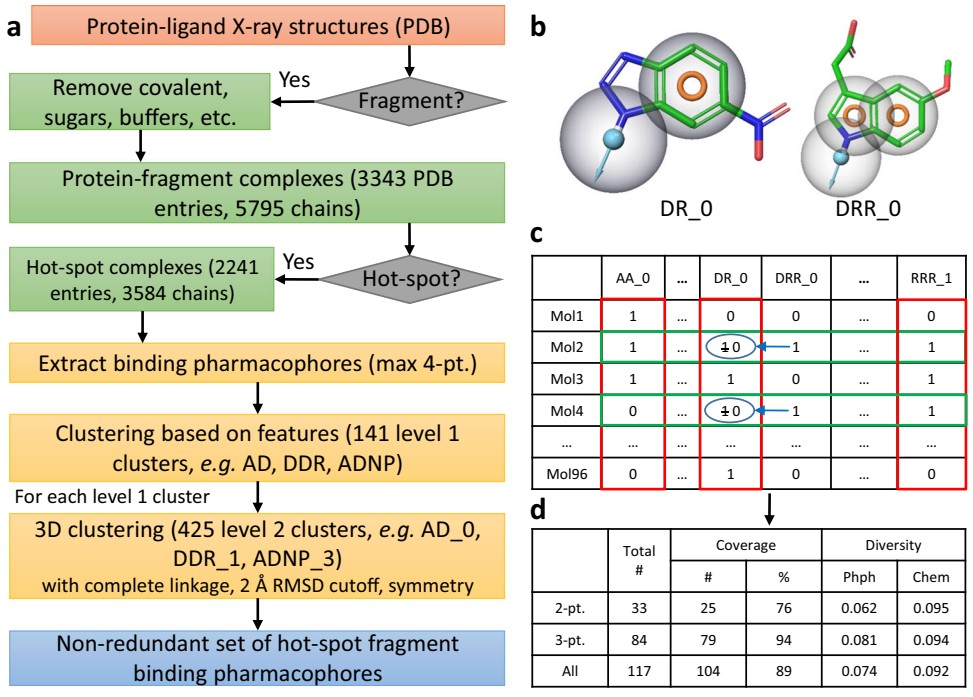

**Fig. 1 SpotXplorer workflow. a** Computational workflow for assembling the non-redundant set of fragment binding pharmacophores. Fragment-sized[15] ligands from the Protein Data Bank (PDB) were subjected to large-scale FTMap[16, 17] analysis to identify hotspot binding fragments. Pharmacophore features (A—H-bond acceptor, D—H-bond donor, H—hydrophobic group, N—negative charge, P—positive charge, R—aromatic ring, see Supplementary Information, section 1.7 for more detail) with the largest contributions to the overall free energy of binding were clustered based on their respective feature sets and binned by paired root-mean-square deviation (RMSD) values (e.g., DRR_0, with 0 being an arbitrarily assigned identifier). **b** Pharmacophore models overlaid onto the co-crystallized representative ligand of the cluster. For each fragment fingerprint generated, fit to a larger model necessarily includes smaller models. Here, DR_0 can be fitted onto the corresponding features of DRR_0; the former is a submodel of the latter. **c** Compound selection algorithm. Iterative minimization of both the mean pairwise fingerprint similarity (green rows) and pharmacophore similarity (red columns) with submodels set to zero and simultaneous maximization of the total number of represented pharmacophores generated the SpotXplorer0 pilot library (96 compounds). **d** Final coverage and distribution data of the 96-compound set selected using 2-point and 3-point pharmacophores (along with the total numbers of non-redundant 2-point and 3-point pharmacophores). Pharmacophore (Phph) and chemical (Chem) diversity is measured as mean Tanimoto similarities[59].

formats, one with all pharmacophores present and a second with submodels absent.

With the post-processed pharmacophore information in hand, we applied a custom-made optimization algorithm for compiling 96 fragment-sized molecules from the filtered vendor datasets. First, 96 molecules were selected by the MaxMin algorithm[23], based on the distances of their pharmacophore fingerprints. Next, we defined an objective function as the sum of the diversity of the selected molecules, the diversity of the pharmacophores (Fig. 1c, red columns and green rows, respectively), and the overall coverage of the pharmacophores (ratio of pharmacophores with at least one matching molecule). Then, members of the selected fragment set were swapped with other molecules, as long as the swap increased the objective function of the algorithm. For the last 10% of compounds, acceptable candidate structures did not significantly alter the overall diversity, so candidates were chosen to fill missing or underrepresented pharmacophores.

Compound purchase from five different vendors yielded the physical SpotXplorer pilot library (SpotXplorer0), containing 96 compounds with a coverage of 76% of the 2-point and 94% of the 3-point, non-redundant binding pharmacophores (Fig. 1d). The library displays a high degree of structural and pharmacophore diversity as calculated from chemical (Chem) and pharmacophore (Phph) fingerprints, respectively. Compared to the collections of the top five most popular commercial fragment vendors identified on the Practical Fragments blog[24], SpotXplorer0 has a significantly higher percentage of compounds that represent unique 2-point and 3-point binding pharmacophores (Supporting Information, section 3).

**SpotXplorer screening yields selective fragment hits for GPCRs and proteases.** The SpotXplorer0 pilot library was first tested in a biochemical screening against the representatives of two popular target classes, G-protein coupled receptors (GPCRs) and proteases. The GPCRs were represented by serotonin (5-HT) receptors, which control key functions in the central and peripheral nervous systems[25]. SpotXplorer0 was tested against three 5-HT receptors: 5-HT$_{1A}$, 5-HT$_6$, and 5-HT$_7$, in a cell-based radioligand binding assay format. Additionally, SpotXplorer0 was tested on the serine proteases Factor Xa and thrombin, which are important molecular targets for anti-coagulant drugs[26,27], using chromogenic protease assays.

Experimental screening successfully identified multiple diverse fragment hits from the SpotXplorer0 library for each of the protein targets (Fig. 2a). Remarkably, there is limited overlap between the hits for the different protein targets. The only exception was the 5-HT$_7$ receptor, known to have similar pharmacophores to 5-HT$_6$[28]. Additionally, the main point of this investigation was to provide a retrospective validation of our library design approach, and more specifically to determine what percentage of the known pharmacophores (represented by previously reported fragment-sized ligands) of a certain protein target was covered by the fragment hits identified by screening the optimized SpotXplorer0 library against the same target.

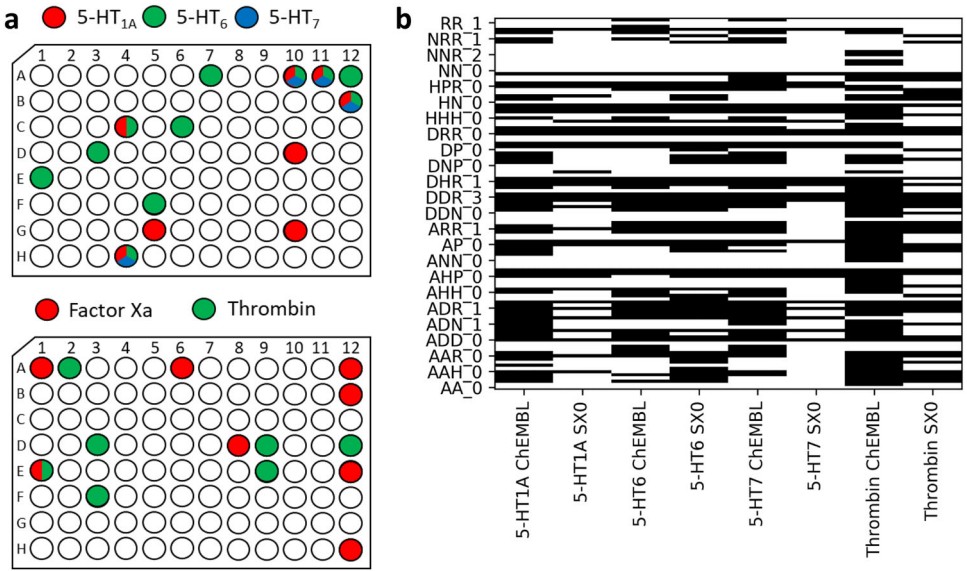

**Fig. 2 Experimental validation of SpotXplorer0 on known target classes. a** SpotXplorer0 pilot library screening yielded a diverse set of confirmed hits for validated GPCR and protease drug targets. Hits are denoted by color in their respective plate positions (upper panel: 5-HT$_{1A}$ with red, 5-HT$_6$ with green, 5-HT$_7$ with blue, lower panel: factor Xa with red, thrombin with green). **b** The handful of hit compounds from SpotXplorer0 represents the majority of pharmacophore models (black) that occur over the respective sets of (typically 4–11 times more) fragment-sized ligands in the ChEMBL database (Percentages are reported in Table 1).

**Table 1 Summary of fragment hits from the SpotXplorer0 pilot library and the ChEMBL database.**

| Target | No. hits (SpotXplorer) | No. fragments (ChEMBL)[a] | Percentage of retrieved pharmacophores | |
|---|---|---|---|---|
| | | | 2-point | 3-point |
| 5-HT$_{1A}$ | 8 | 81 | 80.0 | 51.5 |
| 5-HT$_6$ | 11 | 43 | 100 | 87.5 |
| 5-HT$_7$ | 4 | 44 | 64.3 | 46.4 |
| Factor Xa | 8 | 1 | _[b] | _[b] |
| Thrombin | 7 | 76 | 78.8 | 54.8 |
| Average | – | – | 80.8 | 60.0 |

Retrieval rates of the pharmacophores found in ChEMBL fragments by the SpotXplorer hit compounds are reported as percentages.
[a]Active fragments were defined as molecules with 10–16 heavy atoms displaying at least millimolar activity (expressed as IC$_{50}$, EC$_{50}$, $K_i$, or $K_d$).
[b]Due to the small number of active fragments in ChEMBL, retrieval rates were not calculated.

To estimate the mapping efficiency of the SpotXplorer0 library on the binding hotspots of the individual protein targets, we analyzed the respective sets of reported fragment-sized ligands in the ChEMBL database, considering those fragments with at least millimolar activity expressed as IC$_{50}$, EC$_{50}$, $K_i$ or $K_d$[29]. The respective fragment sets were downloaded, prepared and analyzed with the same workflow that we developed for assembling the SpotXplorer0 library itself (Supplementary information, section 2.1). Then, the list of matching pharmacophores was collected for each fragment; combining these lists for the fragment-sized ligands of a protein target defines the binding pharmacophores utilized by the given target. The analogous pharmacophore lists retrieved by SpotXplorer0 hits were cross-checked against those of the ChEMBL fragment sets to define, what percentage of the known binding pharmacophores of a given target is covered by the respective fragment hits from our library.

Remarkably, while there were up to 11 times more fragments reported in ChEMBL (Table 1), the limited number of fragment

hits from SpotXplorer0 matched the majority of binding pharmacophores present in the ChEMBL sets (most remarkably for 5-HT$_6$, where all 2-point and 87.5% of 3-point pharmacophores were retrieved). Figure 2b provides a visual comparison of the respective pharmacophore sets with black bars corresponding to pharmacophores, that are represented by at least one fragment in the respective ChEMBL sets or SpotXplorer0 hit lists. We can therefore conclude that the minimal set of fragments collected to SpotXplorer0 retrieved most of the binding pharmacophores known from the literature.

In addition to pharmacophore coverage, we compared some basic topological features of the SpotXplorer0 hits vs. ChEMBL fragments (Fig. 3). Information about the hydrogen bonding patterns and general ring types of each pattern is embedded in the pharmacophores. Although there is a wide range of ring types and hydrogen bond patterns in the overall set, each target selected SpotXplorer0 hit sets with rings and hydrogen bonds similar to the distributions of their ChEMBL fragments. This provides further evidence that a well-designed small fragment library can contain the majority of the information required for productive target binding. For the targets studied, fragments containing both aromatic and non-aromatic rings were preferentially identified as experimental hits (as represented by the light green rings in Fig. 3). This is consistent with previously identified preferences for aromatic character in active fragments[9].

Additionally, in the case of Factor Xa and thrombin, the large number of experimental structures available in the PDB database allows for an implicit comparison of fragment binding poses. The possible binding poses of the fragment hits are determined by the available arrangements of their pharmacophore features, since these are the structural moieties forming the critical interactions with the surrounding binding site residues. Therefore, we performed a comprehensive and comparative analysis of X-ray validated binding pharmacophores extracted from all available thrombin and Factor Xa structures, and those pharmacophores identified by screening the SpotXplorer0 library against these targets.

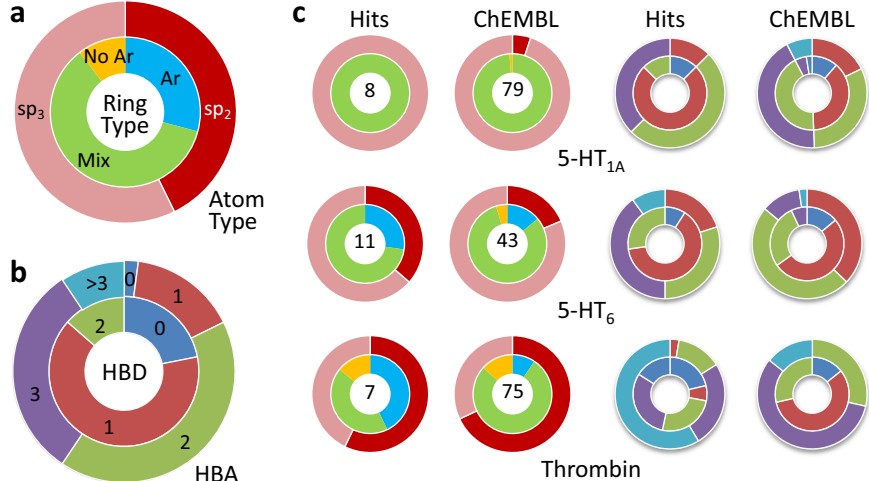

**Fig. 3 Topological features of SpotXplorer0 hits vs. ChEMBL fragments. a** Selection by pharmacophore results in a relatively even distribution of both ring types (Ar: only aromatic—light blue, Mix: both aromatic and non-aromatic—light green, No Ar: non-aromatic—yellow) and both atom types (sp2 hybridization—dark red, vs. sp3 hybridization—light red/salmon). **b** Overall hydrogen bond distribution of ligands in SpotXplorer0 resulting from pharmacophore matching (HBD: no. of H-bond donors, HBA: no. of H-bond acceptors; 0—dark blue, 1—red, 2—light green, 3—purple, >3—light blue). **c** Comparison of SpotXplorer0 hit compounds and ChEMBL fragments for selected targets. Each of the targets shown selects non-overlapping ring subsets: 5-HT$_{1A}$: mixed ring types with high sp$_3$ character and few hydrogen bond sites; 5-HT$_6$: flatter rings, more aromatic character with moderate hydrogen bond site counts; Thrombin: non-aromatic sp$_2$ rings with higher hydrogen bond site counts (Colors are identical to panels **a** and **b**.).

All thrombin and Factor Xa structures were downloaded from the PDB. Apo structures were eliminated and individual chains were identified from the holo structures, resulting in 214 and 118 chains for thrombin and Factor Xa, respectively. These chains were used to identify experimentally validated binding pharmacophores according to the same protocol developed for assembling the non-redundant set of binding pharmacophores (Supplementary information, section 1.2). Out of the 190 and 97 experimental pharmacophores 124 (65.3%) and 72 (74.2%) of the thrombin and Factor Xa pharmacophores were included in the non-redundant set used for designing the SpotXplorer0 library. Focusing to the 58 pharmacophores defined by the fragment-sized ligands of thrombin, these were all retrieved by the non-redundant set.

Next, we crosschecked the individual 2-point and 3-point pharmacophores appearing in the reported thrombin and Factor Xa hits from SpotXplorer0 against those that were identified in the X-ray structures of the respective targets, and found that 86.7% and 85.0% of the X-ray validated pharmacophores were represented by the SpotXplorer0 hits, respectively. Focusing to the 58 thrombin-fragment complexes, 90.0% of the X-ray validated pharmacophores were identified successfully by screening SpotXplorer0.

Finally, we checked whether the screening hits identified pharmacophore arrangements that are not available from the PDB structures. For this analysis we used only thrombin, since there is only one fragment complex of Factor Xa available. Considering the 58 thrombin-fragment complexes, there are six different 2-point and four different 3-point pharmacophores, while the SpotXplorer0 hits for thrombin represent 26 distinct 2-point and 43 distinct 3-point pharmacophores. Consequently, our approach identified 20 and 39 additional 2-point and 3-point pharmacophores, representing potentially novel poses for the fragment screening hits.

**Fragment hits for challenging and current targets**. In addition to the well-established GPCR and protease targets, the SpotXplorer0 pilot library was screened against more challenging current targets, including the histone methyltransferase SETD2, as well as the main protease 3CLPro and the macrodomain NSP3 of the SARS-CoV-2 virus, which is responsible for the ongoing COVID-19 pandemic.

SETD2 is the only mammalian histone methyltransferase that can tri-methylate the side chain of K36 on histone H3[30]. Its role in the progression of acute myeloid leukemia (AML) was established recently, making it a promising oncotarget[14]. So far, only the moderately active natural nucleoside sinefungin and its derivatives were reported as inhibitors of SETD2 and related enzymes[31]. However, they are not cell permeable and lack selectivity for SETD2. Screening the SpotXplorer0 pilot library against SETD2 in a chemiluminescence-based enzymatic assay resulted in two fragments, SX045 and SX084, inhibiting SETD2 function in a dose-dependent manner with IC$_{50}$ values of 300 and 500 μM, respectively (Supplementary Information, section 7).

We investigated the effect of the more potent chemotype (SX045) against SETD2 in a cell-based assay using MOLM-13 and MV4-11 leukemia cells, which were shown before to be particularly sensitive to SETD2 perturbation[14]. SX045 exhibited anti-proliferative effects at an EC$_{50}$ of 333 and 400 μM in MOLM-13 and MV4-11 cells, respectively (Fig. 4a). Even though this fragment was not optimized for cell permeability, this promising result represents a starting point for further drug development.

We are currently experiencing the COVID-19 pandemic that is caused by the SARS-CoV-2 virus[32]. At the time this article is written, there are over 150 million confirmed cases and over 3 million deaths worldwide[33]. The scientific community has rapidly reacted to the unfolding situation, with worldwide collaborative efforts established at an unprecedented pace. Notably, the structure of the SARS-CoV-2 main protease 3CLPro, one of the most promising viral targets, was published in early February 2020[34]. 3CLPro is a viral protease that cleaves the replicated viral polypeptides into functional proteins[35]. Viral protease inhibition is a well-established strategy for the treatment of viral diseases such as HIV, and HIV protease inhibitors have entered into early clinical trials against SARS-CoV-2[36] as well as MERS-CoV[37] (although the lopinavir/ritonavir trial against COVID-19 was discontinued in July 2020 due to ineffectiveness)[38].

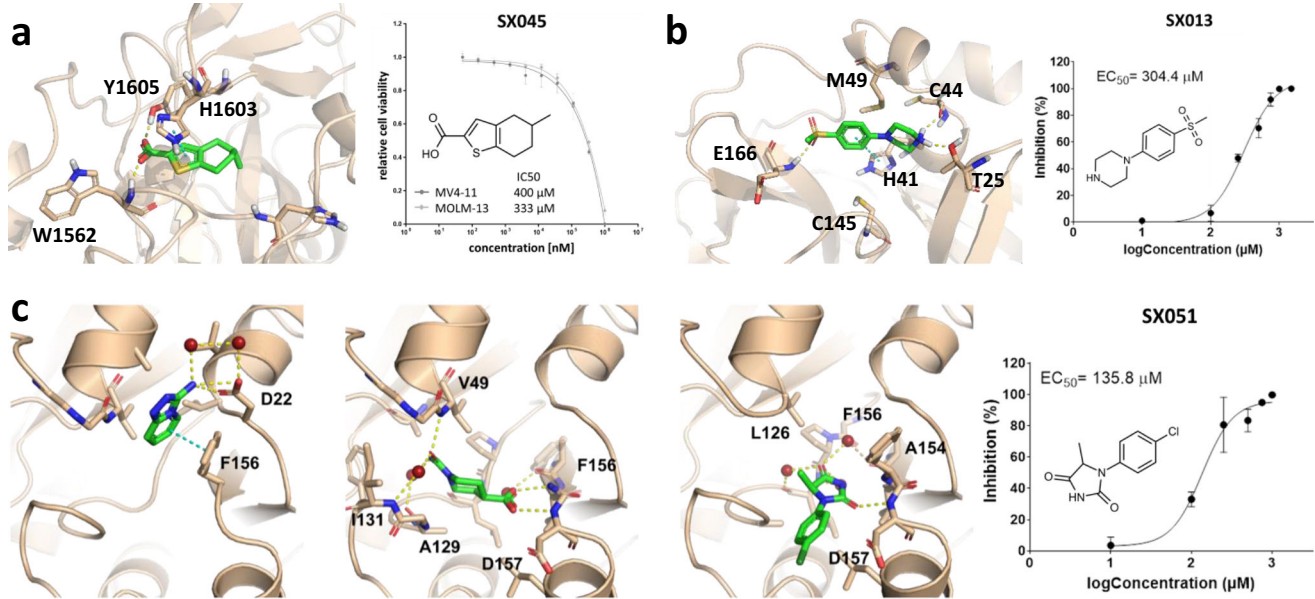

**Fig. 4 Binding poses and cellular activities of SpotXplorer0 hits against challenging and current targets. a** Predicted binding pose of the fragment SX045 (green) in the binding pocket of the SETD2 histone methyltransferase. The fragment decreases the viability of MV4-11 and MOLM-13 leukemia cells in a dose-dependent manner (see Supplementary Information, section 8), with IC$_{50}$ values of 400 µM and 333 µM, respectively. **b** X-ray structure of the fragment SX013 in complex with the main protease (3CLPro) of the SARS-CoV-2 virus (PDB entry 5RHD). The fragment inhibits the SARS-COV-2-induced mortality of Vero E6 cells with an EC$_{50}$ of 304 µM. **c** X-ray structures of fragments SX005, SX048 and SX051 (left to right) in complex with the NSP3 macrodomain of the SARS-CoV-2 virus (PDB entries 5S4G, 5S4H, and 5S4I). The fragments show antiviral activities with EC$_{50}$ values in the high micromolar range in the cellular assay. (In the IC$_{50}$ and EC$_{50}$ plots, data are presented as mean values $+/-$ SD, calculated from $n = 3$ biologically independent samples.) Source data are provided in the Source Data file.

SARS-CoV-2 NSP3 is a multidomain protein harboring a macrodomain module (also referred to as macro X domain) with enzymatic ADP-ribosyl hydrolase activity[39], that has been shown to suppress human interferon response allowing efficient replication of the virus in human cells. Therefore, inhibition of the NSP3 macrodomain has been suggested as a treatment possibility of coronavirus infections, however, no potent inhibitors for this enzyme have been discovered to date[40].

X-ray crystallography is a very sensitive technique for the detection of weak binders such as fragments (with affinities in the high millimolar range), and instantly provides structural information that can be used to progress the fragments into potent compounds[41,42]. With the recent developments at synchrotrons in high-throughput and automation, it is now possible to perform an in-crystal fragment screening campaign within a week. The XChem platform at Diamond Light Source combines a semi-automatic sample preparation facility with a fully automated high-throughput beamline. The entire pipeline involves soaking of the compounds into the crystals[43], their harvesting and unattended data collection. Data analysis and hit identification are managed within XChemExplorer[44], which uses the Pan Dataset Density Analysis (PanDDA) method for the identification of weak ligands[45]. As part of Diamond's contribution to the fight against COVID-19, SARS-CoV-2 3CLPro, and NSP3 macrodomain were screened against SpotXplorer0.

The 3CLPro X-ray screen has resulted in SX013, an aryl-piperazine fragment located in the active site, in the vicinity of the catalytic residue C145. The aromatic ring sits at the entrance of the flexible S2 subsite with the sulfone substituent pointing towards the S3 subsite and the piperazine moiety located near the S1' subsite. The benzene ring provides hydrophobic interactions with the M49 side chain, while the nitrogen of the piperazine moiety forms hydrogen bonds with the C44 carbonyl group and T25 hydroxyl group. The sulfone also interacts with the E166

backbone through hydrogen bonding (Fig. 4b). The location of this fragment, central to all subsites, suggests it could be further expanded into S1 and S1' subsites from the aromatic ring and the piperazine, while on the other side, the sulfone can be conveniently used to reach the S3 pocket. The fragment inhibited 3CLPro enzyme activity with an IC$_{50}$ of 31 µM (Supplementary Information, section 9)[46], and was effective in blocking SARS-CoV-2 replication in the Vero E6 cell line with an EC$_{50}$ of 304 µM.

Furthermore, five SpotXplorer0 fragments were identified in the active site of the NSP3 macrodomain. Comparison of the fragment-bound structures with the macrodomain structure in complex with the natural ligand ADP-ribose (PDB ID: 6WOJ) revealed that three of the fragments (SX003, SX005, and SX054) bound just above the position (towards the solvent phase), which is occupied by the adenine moiety of ADP-ribose. All three fragments also overlay with each other and target the aromatic sidechain of F156 by (stacked) π–π interactions with their aromatic bicyclic scaffold. SX005 additionally interacts by hydrogen bonding with its primary amine with the sidechain of aspartate D22 and a structural water. The other two identified fragments, SX048 and SX051, also bind within the adenosine-binding pocket, however, they overlay in position with the adenine-proximal ribose of the ADP-ribose. While SX051 seems to undergo interaction only with its imidazolidine-2,4-dione moiety by forming a hydrogen bond with the backbone NH of aspartate D157, SX048 is held nicely in place with two functional groups being in positions 1 and 4 of the piperidine scaffold. Thus, SX048 interacts with its primary amide with the V49 backbone and its carboxyl group allows targeting both backbone NH's of F156 and D157 by hydrogen bonding. Furthermore, both functional groups engage with the water network in the binding pocket. Significant antiviral activities were detected for SX005, SX048 (Supplementary Information, section 10), and SX051

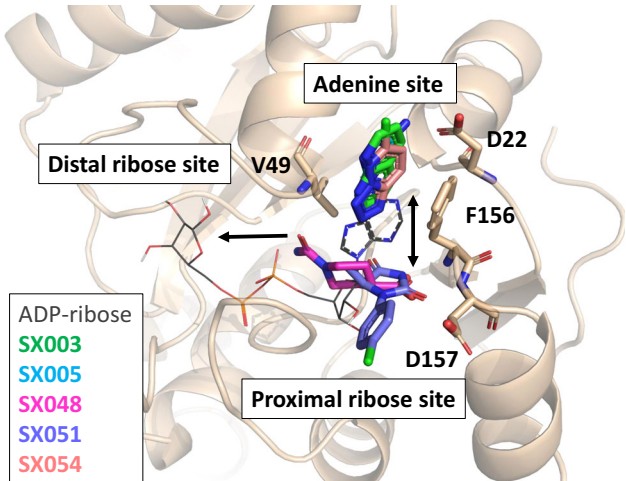

**Fig. 5 SpotXplorer0 hits against the SARS-CoV-2 NSP3 macrodomain.** Overlay of the binding poses of the five SpotXplorer0 hits against the SARS-CoV-2 NSP3 macrodomain (colored sticks, SX003—green, SX005—light blue, SX048—magenta, SX051—purple, SX054—orange) with the binding pose of ADP-ribose (gray lines, from PDB structure 6WOJ[60]). Three and two hits occupy the adenine and proximal ribose sites, respectively, providing merging and growing options towards the neighboring subsites (indicated by black arrows).

(Fig. 4c) with $EC_{50}$ values as low as 136 μM (for SX051) in our in vitro infection assay.

## Discussion

Fragments have been established as a compact representation of a much larger chemical space, making the use of fragment libraries an attractive approach for primary hit discovery[47]. Pharmacophores are the most generalized representative model for the spatial layout of physical features required for protein–ligand binding. This allows for a very good performance-to-cost ratio[48,49], making pharmacophore-based computational screening a popular method in drug discovery[50]. To further decrease computational cost, pharmacophores can be readily implemented into fingerprint-based approaches[51,52]. Since this does not place any constraints on the actual chemical composition, the pharmacophore concept is especially well-suited for tasks like scaffold hopping[53,54] or library design[55].

Compared to earlier approaches, the key difference in our SpotXplorer library design stems from our realization that the practical pharmacophore space is unevenly represented. Binding hotspots are evolutionarily conserved and contribute the majority of binding free energy for any ligand. The small fraction of fragments that bind hotspots with a few high-value interactions are generally privileged while the remainder are rarely active[9]. Consequently, instead of systemically generating a combinatorial pharmacophore ensemble[55], we have collected and analyzed the publicly available set of experimental protein-fragment complexes in the PDB. A non-redundant set of 425 pharmacophores with a maximum of four features covers nearly all the described target–ligand interactions. By optimizing a fragment library to cover the experimentally verified pharmacophore set in a diverse way, i.e., where each pharmacophore was represented by a diverse set of fragments, we aimed to maximize our chances for hit discovery. This approach could provide viable starting points against virtually any available protein target.

Recognizing the value of minimal pharmacophores to fragment binding[15], we have produced a 2-point and 3-point pharmacophore optimized library of 96 fragments, the SpotXplorer0 pilot

library. To measure the redundancy present within our selection (in addition to Fig. 1d), we have performed a principal component analysis on the 425 hotspot pharmacophore fingerprints, without submodels, of SpotXplorer0. We have found that only 45% of the overall data variance can be explained by the first ten principal components, providing good evidence that the occurrences of a given pharmacophore are largely independent from the occurrences of others. By comparison, in highly correlated datasets like molecular dynamics simulations of macromolecular systems or sample classification based on UV/Vis- or infrared spectra, the first ten principal components often explain almost 100% of the overall variance. Even when the necessarily correlated submodels are included in the dataset, the explained variance of the first ten principal components rises only to 58%. Additionally, we have compiled key information on the occurrence of certain pharmacophoric features (polar, directional, and apolar features, etc.) in the Supplementary Information, section 1.6. Notably, there are only 18/6/1% of 2/3/4-pt. pharmacophores (3% in total) that do not contain any polar features such as H-bond acceptors/donors or ionic centers. This is in line with our previous observation that apolar desolvation of fragments alone cannot compensate for rigid-body entropy loss upon binding[5].

In contrast to the conventional ligand-based library design strategies, SpotXplorer is a protein-based approach that highlights the importance of explicitly accounting for the experimental binding pharmacophores in available fragment-protein structures. SpotXplorer0 shows an excellent coverage of the experimentally validated pharmacophore space (76% and 94% for unique 2-pt. and 3-pt. pharmacophores, respectively). By comparison, a recently published fragment library of the same size that was compiled by ligand-based methods (ROCS shape and pharmacophore scores) provides significantly lower coverages (30% and 49% for unique 2-pt. and 3-pt. pharmacophores, respectively)[56]. On the other hand, moving from a ligand-based to a more protein-based approach might introduce other potential limitations, such as the correct identification of protonation states, tautomeric forms and protein-induced polarization effects. To tackle this challenge, our protocol assigns hydrogen positions by taking into account relevant $pK_a$ values and possible H-bonds, assigns charges and predicts the most reliable tautomer of the ligand. This represents a feasible option to consider these effects as efficiently as possible for the large number of experimental protein-fragment complexes (close to 4000) used in our study (Supplementary Information, section 1.2). Also, while the current set of non-redundant binding pharmacophores is inherently limited to those that have appeared in at least one publicly available experimental structure in the PDB, it can be updated to include newly released structures. Alternatively, the same workflow can be utilized to extract a non-redundant set from a different source of experimental structures (such as proprietary in-house databases).

Experimental screening against several well-established protein targets has validated our hypothesis, providing a diverse set of fragment hits for three GPCRs and two proteases (Fig. 2). Additionally, our approach has successfully yielded fragment hits for novel, challenging targets such as the histone methyltransferase SETD2 and two potential drug targets of the SARS-CoV-2 virus, the main protease 3CLPro and the NSP3 macrodomain. In the latter two cases, X-ray crystallography was employed as the primary screening format, providing direct feedback and validation of our pharmacophore-based approach. The experimental complex of SARS-CoV-2 3CLPro with the fragment SX013 documents PR_0 as the binding pharmacophore, which was indeed a match to SX013 during pharmacophore screening (see Supplementary Data).

Screening the SpotXplorer0 library against the NSP3 macro-domain as a novel coronavirus target provided even more proof of concept. The identified pharmacophores can be classified into the three clusters DR_0, AN_1 and AAR_0 with different underlying chemical scaffolds, which engage with several sites of the ADP-ribose pocket. Importantly, the identified fragments mimic interactions of the natural ligand ADP-ribose as well as establish additional interactions with the macrodomain (Fig. 5). The three fragments of the first cluster (SX003, SX005, and SX054) are characterized by aromatic bicycles that target the F156 sidechain—with the H-donor in SX005 additionally targeting the sidechain of D22, which resemble the interactions of the adenine moiety of ADP-ribose with the macrodomain. With its primary amide group, SX048 is able to target V49 which naturally interacts with the phosphate group of the ADP-ribose. Furthermore, fragment SX048 establishes with its carboxyl group new interactions with the macrodomain, which are different compared to ADP-ribose. This hit provides valuable hints of targetable regions for inhibitor development and presents a candidate for merging with other fragments that present more traditional binding poses, especially since its antiviral effect was already demonstrated in an in vitro infection assay. Moreover, both SX048 and SX051 provide growing vectors towards the distal ribose site (with their amide and methyl groups, respectively), enabling their optimization into highly specific ligands.

Overall, we found that pharmacophore optimized fragments are mapping the binding sites of protein targets effectively. This approach was capable of retrieving fragment hits by matching the known and validated pharmacophores of well-established targets, thereby providing chemical starting points for novel and challenging proteins.

## Methods

**Clustering of experimental structures to non-redundant set of binding pharmacophores**. Fragment-sized ligands of 10–16 heavy atoms[15] were filtered to exclude covalent labels, sugars, buffers, and crystallization additives, and subjected to large-scale FTMap[16,17] analysis to identify fragments bound to hotspots. Schrödinger's ePharmacophore protocol was employed to extract pharmacophore models containing a maximum of four features (A—H-bond acceptor, D—H-bond donor, H—hydrophobic group, N—negative charge, P—positive charge, R—aromatic ring) with the most significant contributions to the overall free energy of binding, approximated by the Glide XP docking score (without changing the binding mode)[18,19]. The observed pharmacophore models were clustered based on their respective feature sets (level 1, e.g., DRR—an H-bond donor and two aromatic rings). Then the models of each level 1 cluster were spatially aligned and clustered based on their root-mean-squared distance (RMSD) values (level 2, e.g., DRR_0, with 0 being an arbitrarily assigned identifier of the specific 3D layout of the pharmacophoric features). The workflow resulted in 425 unique level 2 clusters. For each level 2 cluster, the pharmacophore model that is closest to the cluster centroid was selected as the cluster representative.

**Fragment library optimization**. Vendor fragment sets were filtered for size, rotatable bond count and other properties, and the absence of known problematic features[22]. Candidate fragments were annotated with the full 425 non-redundant set of 2, 3, and 4-point pharmacophore models and stored as fingerprints with bits for pharmacophores present set to 1. When a molecule fits a 4-point (or 3-point) model, it trivially fits 2-point and 3-point models with the same geometry (sub-models). To ensure that these pharmacophore models are represented on their own rather than only as part of a larger model, a second fingerprint was generated with bit positions for submodels set to zero (see Fig. 1c, blue circles).

The desired number of molecules was compiled with an algorithm that iteratively and simultaneously minimized the mean pairwise fingerprint similarity of the selected molecules (Fig. 1c, green rows), and pharmacophores (Fig. 1c, red columns), and maximized the number of models represented by at least one molecule (coverage). For the SpotXplorer0 pilot library, we selected 96 compounds focusing on the set of 2-point and 3-point pharmacophores with the 2-point submodels removed. For the last 10% of compounds, acceptable candidate structures did not significantly alter the overall diversity, so candidates were chosen to fill missing or underrepresented pharmacophores.

**Biochemical screening against the protein targets**. GPCR radioligand binding assays were performed in HEK293 cells with stable expression of human 5-HT$_{1A}$,

5-HT$_6$, and 5-HT$_{7b}$ receptors (prepared with the use of Lipofectamine 2000). Non-specific binding was defined with 10 μM of 5-HT in 5-HT$_{1A}$R and 5-HT$_7$R binding experiments, whereas 10 μM of mianserin was used in 5-HT$_6$R assays, respectively. Each compound was tested at 10 μM concentration. Results were expressed as means of at least two separate experiments. Fragments that exhibited 50% or stronger inhibition were considered as hits.

Protease inhibitory assays were performed in transparent microtiter plates in a final volume of 200 μL with the use of chromogenic substrates. The reaction rates in the absence and in the presence of the inhibitor were measured; screenings were carried out in duplicate in one independent experiment and $K_i$ values were determined in triplicate in two independent experiments.

The effect of SX045 on the viability of MOLM-13 and MV4-11 leukemia cells was evaluated using the CellTiter-Glo Luminescent assay. Cell viability was determined 5 days after treatment and IC$_{50}$ values were determined using serial dilutions starting from 1 mM in biological triplicates.

Cellular SARS-COV-2 inhibitory activities were determined in infected Vero E6 cells. Viral copy numbers were determined 48 h post infection, using SARS-CoV-2 RdRp gene specific primers (listed in Supplementary Table 2) and droplet digital PCR (Bio-Rad Laboratories Inc. QX200 Droplet Digital PCR System).

**X-ray screening against SARS-Cov-2 main protease (3CLPro) and NSP3**. Crystals of the 3CLPro and NSP3 macrodomain *apo* proteins were grown using the sitting drop vapor diffusion method at 20 °C. NSP3 macrodomain *apo* crystals were grown in crystallization drops containing 150 nl of protein solution (47 mg/ml in 20 mM HEPES pH 8.0, 250 mM NaCl, and 2 mM DTT) plus 150 nl crystallization solution (100 mM CHES pH 9.5 and 30% PEG3000)[57]. For 3CLPro, crystallization drops contained 150 nl protein solution (5 mg/ml in 20 mM HEPES pH 7.5 and 50 mM NaCl), 300 nl crystallization solution (11% PEG 4K, 5% DMSO, 0.1 M MES pH 6.7) and 50 nl seeds[58]. For both SARS-CoV-2 main protease (3CLPro) and NSP3 macrodomain, fragments were soaked into crystals by acoustic dispensing[43], adding dissolved compound directly to the crystallization drops using an ECHO liquid handler (final concentration 10% DMSO); drops were incubated for approximately 1–3 h prior to mounting and flash freezing in liquid nitrogen.

Data were collected at the beamline I04-1 at 100 K and automatically processed with Diamond Light Source's auto-processing pipelines. Most SARS-Cov-2 main protease (3CLPro) data processed to a resolution of approximately 1.8 Å and NSP3 macrodomain to 1.1 Å. For both targets, data with resolution below 2.8 Å were excluded. Coordinates, structure factors and PanDDA[45] event maps for the structures discussed are deposited in the Protein Data Bank (PDB IDs 5RHD, 5S4F, 5S4G, 5S4H, 5S4I, and 5S4J). Data collection and refinement statistics are summarized in the Supplementary Data file, while a more detailed method description is included in section 9 of the Supplementary Information.

**Reporting summary**. Further information on research design is available in the Nature Research Reporting Summary linked to this article.

## Data availability
The SpotXplorer0 library is available for screening at Diamond Light Source (https://www.diamond.ac.uk/Instruments/Mx/Fragment-Screening/Fragment-Libraries0.html). Structure data that support the findings of this study have been deposited in the PDB database (https://rcsb.org), with the accession codes 5RHD, 5S4F, 5S4G, 5S4H, 5S4I, and 5S4J. Data generated during the computational and experimental screening of the described fragment library are reported in Supplementary Data 1. Structures for assembling the set of non-redundant pharmacophores were downloaded from the Protein Data Bank (https://www.rcsb.org/). GPCR and protease ligands and bioactivity data were downloaded from the ChEMBL database (https://www.ebi.ac.uk/chembl/). Source data are provided with this paper.

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

## Acknowledgements

The authors thank the COVID Moonshot collaboration, particularly Anthony Aimon and Tamás Szommer (Diamond Light Source), Lizbé Koekemoer (University of Oxford, Structural Genomics Consortium), Tika Malla and Anthony Tumber (Department of

Chemistry, University of Oxford, Schofield group) for their contributions in the experimental characterization of hit compounds. G.M.K. and D.B. are grateful to Péter Pogány, Darren Green and Mike Hann (GlaxoSmithKline, UK) for early discussions on some of the ideas in the paper. D.B. is grateful to Anita Rácz for helpful discussions. D.B. was supported by the János Bolyai Research Scholarship of the Hungarian Academy of Sciences. J.E. was supported by grant no. 857935 from the Austrian Research Promotion Agency (FFG). F.G. has received funding from the European Research Council (ERC) under the European Union's Horizon 2020 research and innovation programme (grant agreement no. 636855). J.I. was supported by the Slovenian Research Agency (Grant P1-0208). G.M.K. received funding from the National Brain Research Found of NKFIH, Hungary (Grant NAP 2.0) and the Foreign Commonwealth and Development Office (UK). F.J. and H.P. were supported by the Hungarian Scientific Research Fund (OTKA KH129599), by the European Union and the European Social Fund (EFOP-3.6.1.-16-2016-00004), and by the Ministry for Innovation and Technology of Hungary (TUDFO/47138/2019-ITM).

## Author contributions

D.B. and G.M.K. conceived the study and developed the SpotXplorer approach, with advice from W.S.W., J.G., P.P. and S.V. D.B. and A.W. performed theoretical calculations, with advice from W.S.W., S.V. and G.M.K. P.P. provided materials for the SpotXplorer0 library. G.S., A.J.B., J.I., J.E., F.G., H.P., F.J., A.D., D.F., F.v.D., M.S. and I.A. contributed to experimental design and methodology. G.S., J.I., J.E., H.P., A.D., D.F. and M.S. performed experimental work. All authors contributed to writing and reviewing the manuscript, with leading contributions from D.B. and G.M.K.

## Competing interests

Péter Pallai is the CEO and owner of Bioblocks. All remaining authors declare no competing interests.
