## [Peer Review File · Nature Communications]

Reviewers' Comments:

Reviewer #1:

Remarks to the Author:

The paper of Bajusz et al. presents a new fragment library, named SpotXplorer0, compiled by criteria based on pharmacophore patterns observed in known crystal structures as extracted from the pdb. The library has been tested on three GPCRs, two proteases and two novel targets of present research, the methyltransferase SETD2 and the NSP3 macrodomain of SARS-CoV-2. The paper has potential to be accepted for publication, however, at several points the current manuscript is rather weak and unclear with respect to the documentation of the methods and results.

1. The idea to go through known crystal structures of fragment complexes and to seek for pharmacophores that cover and match with the hot spots indicated by the FTmap method is interesting as it moves the library design away from a sole ligand-based to a more strongly protein-based approach. Criteria such as "what the protein really requires" are supposedly much better covered by this strategy. However, such a procedure also contains some deficiencies and limitations that the authors are requested to mention in their paper. First of all, an algorithm-based assignment of pharmacophore patterns to the molecular scaffold of fragments bears the danger that important features are not considered, e.g. the correct protonation state of functional groups at the binding site which can easily turn an H-bond donor into an acceptor and vice versa, particularly as such effects will depend on the local protein environment. The same holds for tautomeric forms and other protein-induced polarization effects which can alter the properties particularly of aromatic compounds. Secondly, an approach which is based on database information expects that features are exhaustively seen in the considered data sample, otherwise such an approach can easily generate "self-consistency" in an incomplete selection of a fragment sample. Only that what has been observed in the past will be allowed to enter again into the selection of the new pharmacophore-filtered fragments. This is definitely a disadvantage of a protein-based approach compared to a selection which is only based on the comparison and features of the ligands. This has to be mentioned in the discussion.
2. In the analysis of the pdb structures, it is not clear whether the considered pharmacophores are requested or analyzed whether they interact with a similar local protein environment. This might be important for the avoidance of too much redundancy in the sample collection.
3. Perhaps for the unexperienced reader the authors should explain by two or three sentences what the FTMap method does and which assumptions it contains. Furthermore, what is a 2-, 3-, or 4-point pharmacophore, mainly whether a hydroxyl group, simultaneously operating as a donor and acceptor, is a 1- or 2-point pharmacophore and a carboxamide groups, interacting in the same way with the protein, is a 2-point pharmacophore? These definitions must be clear-cut for the reader. A short summary of the nomenclature used to specify the pharmacophores will help (e.g. D, A, R are clear, but what is "N"?).
4. Some fragments contain pharmacophoric points of the same feature quality, e. g. DD. In the real case, which one will win over the other? In the analysis, the authors must have taken such a decision, however, nothing is disclosed in the paper.
5. With the Schrödinger software the authors extracted a pharmacophore model for each protein-ligand complex. Then, they anticipated "a maximum of four available features having the largest energetic contributions to protein-ligand binding". This selection is a crucial one, thus a better documentation is required. What determines this selection? Particularly, desolvation of a site takes major impact on the energetic contribution of a formed interaction. Has this been considered in the selection?
6. The SpotXplorer0 library has been tested on five known cases and the fragment selection has been performed based on a biochemical assay. Fragments are often difficult to detect due to their weak binding. Therefore, what was the sensitivity level of the applied assays? Some information has to be provided. For the two proteases a fluorogenic substrate, which gives a more sensitive assay, might have provided a larger number of hits.
7. The part on the comparison with the binding pharmacophores present in ChEMBL is hardly comprehensible with the given information. The authors are requested to reformulate this part and to provide more insights in the analysis.
8. The authors mention that aromatic fragments are preferentially selected. The question is whether this observation reflects a "real" preference or results from the above-mentioned "self-consistency". Fragments with aromatic groups have been selected in the past due to their easy

chemical access. Is this aspect finally reproduced due to a biased pharmacophore preselection? At least the reader can expect some critical discussion by the authors.

9. The analysis of the GPCR and protease data remains not really conclusive. How novel are the obtained fragments poses? Without a crystal structure this essential question remains unanswered. The best library suggests a number of fragments to the user which indicate deviating binding poses in a protein environment with different orientations of exit vectors for synthesis. Also fragments with optimal pharmacophore distributions do not help much further if overlapping binding modes reproducing the same interaction patterns are produced. It is highly suggested to the authors to add another crystallographic screen for at least one or two of the targets. Particularly the two proteases Factor Xa and Thrombin are well established and could be easily screened with the facilities present at Diamond within very short time. This extension of the current study is highly suggested to the authors and the journal's editor.

10. The 3CLPro and NSP3 study underlines the importance of performing a crystallographic validation of the ScoreXplore0 library and may disclose its advantage over other libraries as it seems that the found entries suggest alternative binding features even though they bind in an overlapping region of the binding site.

11. It is surprising the authors do not provide, as in the other cases, quantitative assay results for the ligands of the NSP3 macrodomain and only describe that they "show values in the high micromolar range". The missing information should be provided.

12. The chemical formulas of the entries of the fragment library have to be documented at least in the Supporting Information. This is definitely a prerequisite for acceptance of this contribution as only then it is in accordance with good scientific practice.

13. The citations are somewhat biased to the publishing authors. Unfortunately, the Nature series does not allow for the full author list. In the current case, the authors sometimes provide a full author's list, sometimes they shortcut to the first author, which usually is the one who did the work but is the unknown PhD student of a group. Without retrieving the full information from a library database, it remains unclear to the reader from which group a particular contribution originated. This is a rather unfortunate service to the scientific community by the journal. With the above-described revisions the paper should be reconsidered for publication in Nature Communication.

Reviewer #2:

Remarks to the Author:

It has long been known that some endogenous and exogenous molecules target several proteins. Drug discovery has taken advantage of this phenomenon through the development of privileged scaffolds, or the repositioning of drugs. Analysis of the 3D-structures in the Protein Data Bank suggested physical basis of molecular promiscuity. Many fragments have versatile binding properties, as do some druglike ligands. In addition, a limited number of protein pockets can cover the pocketome.

Bajusz et al. skillfully exploited the knowledge about protein/fragment recognition and issued a chemical library which is, to my knowledge, completely original.

The three major assets of this chemical library are:

- Its small size (96 fragments): evaluation of the biological activity of compounds is possible by low throughput biophysical methods
- Its universality: it is suitable for all kinds of target protein
- Its annotation: the pharmacophore models covered by each fragment are indicated

The manuscript first introduces the library design, then demonstrates its usefulness in screening exercises. The manuscript is well written. The message delivered is clear and the results remarkable. The authors have also provided the exact constitution of the chemical library. It is therefore possible to purchase the fragments from suppliers.

****About the method for designing the library**** The authors used proven methods, intelligently combined. I only have minor questions and suggestions:

- Were there requirements about a minimal number of polar / directional probes in pharmacophore models?
- About FTMap, did the authors consider all predicted hotspots, or only the hotspots in clusters? How FTMap results were used?
- Pharmacophore models are in 3D. What is the maximal number of conformers per fragment? How many conformers of a fragment have to match a pharmacophore model to conclude that the fragment matches this pharmacophore model?
- Reference 24: Dr. Erlanson's blog is an excellent source of information although unconventional (not peer reviewed). In the case it is not kept online, I suggest to name the 5 chemical vendors in the article.

****About the screening hits**** The library was designed to cover the maximum number of "true" pharmacophore models with a minimum number of fragments. Therefore, it is suitable for identifying hits at any protein site. The proof of concept was made on eight proteins of different class and biological functions. Five of them are widely studied targets, and therefore could be viewed as "easy". Noteworthy, the authors also worked on three original targets (including two SARS-COV-2 proteins). Active fragments were obtained for all the proteins. The results are very impressive, given the small size of the chemical library.

Different experimental assays were used (cell-based binding, activity of recombinant enzyme, Xray crystallography). Biochemical and functional assays have a high propensity to generate false positive, especially when testing compounds at high concentration (here from 10 micromolar to >500 micromolar). Was this point addressed by a confirmatory assay? Could the authors comment on the validation of hits?

Minor: In July, the WHO announced that it has discontinued trials of the HIV drugs as a treatment for COVID19 patients.

Last, could the authors comment on the solubility of the 96 fragments in water?

Reviewer #3:

Remarks to the Author:

This is a very important manuscript for the drug discovery community for which a substantial part of the novel chemical matter identified across all targets originates from fragment approaches. The manuscript describes an elegant selection and construction of a library of fragments that represents binders of common pharmacophores in proteins. The selection of this 96-member fragment library enabled the successful identification of fragments against two established protein targets (GPCR, protease) as well as with three, more difficult targets (human SETD2, 3CLPro and NSP3 from Sars-CoV2). Especially the identification of initial chemical matter binding to the latter two targets is a very valuable confirmation of the performance of this new fragment library selection.

This paper - and the new fragment library - will be of tremendous use to the fragment-based drug discovery community as it allows with high reliability the identification of fragment hits from a very small collection of just 96 molecules.

Reviewer #1 (Remarks to the Author):

The paper of Bajusz et al. presents a new fragment library, named SpotXplore0, compiled by criteria based on pharmacophore patterns observed in known crystal structures as extracted from the pdb. The library has been tested on three GPCRs, two proteases and two novel targets of present research, the methyltransferase SETD2 and the NSP3 macrodomain of SARS-CoV-2. The paper has potential to be accepted for publication, however, at several points the current manuscript is rather weak and unclear with respect to the documentation of the methods and results.

Thank you for the generally positive evaluation of our work and for the constructive suggestions.

1. The idea to go through known crystal structures of fragment complexes and to seek for pharmacophores that cover and match with the hot spots indicated by the FTmap method is interesting as it moves the library design away from a sole ligand-based to a more strongly protein-based approach. Criteria such as “what the protein really requires” are supposedly much better covered by this strategy. However, such a procedure also contains some deficiencies and limitations that the authors are requested to mention in their paper. First of all, an algorithm-based assignment of pharmacophore patterns to the molecular scaffold of fragments bares the danger that important features are not considered, e.g. the correct protonation state of functional groups at the binding site which can easily turn an H-bond donor into an acceptor and vice versa, particularly as such effects will depend on the local protein environment. The same holds for tautomeric forms and other protein-induced polarization effects which can alter the properties particularly of aromatic compounds. Secondly, an approach which is based on database information expects that features are exhaustively seen in the considered data sample, otherwise such an approach can easily generate “self-consistency” in an incomplete selection of a fragment sample. Only that what has been observed in the past will be allowed to enter again into the selection of the new pharmacophore-filtered fragments. This is definitely a disadvantage of a protein-based approach compared to a selection which is only based on the comparison and features of the ligands. This has to be mentioned in the discussion.

The reviewer first mentioned the consideration of protomers and tautomers as a possible limitation of the protein pharmacophore based approach applied here. In general, we agree that protomeric and tautomeric states have a major impact on the interaction pattern realized in the recognition of the fragments. Therefore, we applied a straightforward strategy to consider these effects. The assignment of the pharmacophore patterns was done by Schrödinger's ePharmacophore module. In this procedure we started from the X-ray structure of the corresponding protein-fragment complex. Before the calculations, the protonation and tautomer states were assigned using the Protein Preparation module of Schrödinger [Sastry, G. M., Adzhigirey, M., Day, T., Annabhimoju, R. & Sherman, W. Protein and ligand preparation: parameters, protocols, and influence on virtual screening enrichments. *J. Comput. Aided. Mol. Des.* **27**, 221–34, 2013]. This protocol assigns hydrogen positions by taking into account relevant pK_a values and possible H-bonds, assigns charges and predicts the most reliable tautomer of the ligand. Next, we run the ePharmacophore module that generates the pharmacophore pattern based on the experimental binding pose. To do this, the approach

identifies and calculates the energetics of the interactions formed between the fragment and the binding site using the Glide docking score without changing the binding pose. Analyzing the contributions of the individual ligand features, ePharmacophore finally identifies the key features that are combined in the output pharmacophore pattern. Considering the large set of protein-fragment complexes investigated here, this protocol is a feasible option to deal with the effect of protonation, tautomerization and polarization.

To the reviewer's second point, we agree that the present work is based on the current set of protein-fragment complex publicly available in the PDB. The methodology, however, could be applied on other datasets generated *e.g.* in a corporate environment. Furthermore, the set of experimental fragment pharmacophores can be generated for the upcoming releases of the PDB using the same protocol described here. Although the present selection of the fragment pharmacophores is definitely biased by the available structures, one can update the set of extracted pharmacophores running ePharmacophores automatically on the updated PDB.

To reflect on these questions in the manuscript, we have updated a few of details in the *Methods/Clustering of experimental structures to non-redundant set of binding pharmacophores* subsection and the Supplementary Information, section 1.2, and added a brief summary into the Discussion, paragraph 4.

2. In the analysis of the pdb structures, it is not clear whether the considered pharmacophores are requested or analyzed whether they interact with a similar local protein environment. This might be important for the avoidance of too much redundancy in the sample collection.

Pharmacophores were extracted from all the protein-fragment complexes available in the present PDB release. Our objective was to identify the non-redundant set of binding pharmacophores. Therefore, the complete set of the extracted pharmacophores were clustered. First level clustering considered only the features, while the spatial arrangements were considered at the second stage. Technical details of the clustering protocol can be seen in the Supplementary Information (section 1.4). We believe that this procedure allowed us to remove the redundancy from the final pharmacophore set.

Nevertheless, the comment of the reviewer prompted us to analyze the abundance of specific pharmacophores in the set of protein-fragment complexes used for this analysis. The result is shown in the Supplementary Information, section 3, which summarizes the number of PDB structures and unique proteins for the twenty most "popular" binding pharmacophores, identified by PDB IDs and Uniprot IDs, respectively. For example, the most frequent pharmacophore *NR_0* contains a negative ionic center (*N*) and an aromatic ring (*R*) in a specific spatial arrangement (labeled by the number 0), and it occurs in 106 PDB structures of 71 unique proteins; yet it is represented by only a single pharmacophore model in the non-redundant set of 425 binding pharmacophores. This demonstrates the power of two-level clustering to identify the unique set of experimentally validated binding pharmacophores.

3. Perhaps for the unexperienced reader the authors should explain by two or three sentences what the FTMap method does and which assumptions it contains. Furthermore, what is a 2-, 3-, or 4-point pharmacophore, mainly whether a hydroxyl group, simultaneously operating as a donor and acceptor, is a 1- or 2-point pharmacophore and a carboxamide groups, interacting in the same way with the protein, is a 2-point pharmacophore? These

definitions must be clear-cut for the reader. A short summary of the nomenclature used to specify the pharmacophores will help (e.g. D, A, R are clear, but what is “N”?).

We are grateful for the comment of the referee that helps the wider community to follow our discussion easier. The theoretical background of FTMap is described in detail in the Supplementary information, and now we added a brief summary to the main text (Results, 1st subsection, 1st paragraph).

Pharmacophore definitions including the nomenclature are now discussed as a separate section of the Supplementary Information (section 1.7, covering also the questions in this reviewer comment). We refer to this in the caption of Figure 1, which also contains a brief note on pharmacophore labels (A,D,H...). This information might help non-specialist readers to follow the main text.

4. Some fragments contain pharmacophoric points of the same feature quality, e. g. DD. In the real case, which one will win over the other? In the analysis, the authors must have taken such a decision, however, nothing is disclosed in the paper.

This is a relevant point and actually, both features should be matched. Fragments having identical features define a symmetric pharmacophore, matching compounds with more than one group acting as pharmacophoric features of the same type. This is explained in the newly added section 1.7 of the Supplementary Information and can be further illustrated with the classical example of ATP site kinase inhibitors, where two donors (corresponding to two separate chemical moieties) are featured in the well-characterized donor-acceptor-donor hinge binding pattern.

5. With the Schrödinger software the authors extracted a pharmacophore model for each protein-ligand complex. Then, they anticipated “a maximum of four available features having the largest energetic contributions to protein-ligand binding”. This selection is a crucial one, thus a better documentation is required. What determines this selection? Particularly, desolvation of a site takes major impact on the energetic contribution of a formed interaction. Has this been considered in the selection?

Major factors considered here are fragment properties and analysis of fragment interactions in protein-fragment complexes. In one hand, fragments typically fit to the Rule of Three that includes simple physicochemical guidelines for compiling fragment libraries. MW should be less than 300 Da, the number of H-bond donors and acceptors should be <3, respectively and logP should be lower than 3. These restrictions suggest that the number of pharmacophore features in fragments should generally not be larger than 6. Limiting the number of features to four agrees with the concept of binding pharmacophores as the smallest set of key pharmacophoric features, with the largest contributions to fragment binding [Murray, C. W. & Rees, D. C. Opportunity Knocks: Organic Chemistry for Fragment-Based Drug Discovery (FBDD). *Angew. Chemie Int. Ed.* **55**, 488–492, 2016]. This was updated in the Supplementary Information, section 1.2 (bullet point vi).

Analyzing the confirmed interactions in protein-fragment complexes revealed that fragments typically form 2 high quality H-bonds [Ferenczy, G. G. & Keserű, G. M. Thermodynamics of Fragment Binding. *J. Chem. Inf. Model.* **52**, 1039–1045, 2012], this was reinforced recently [Giordanetto, F., Jin, C., Willmore, L., Feher, M. & Shaw, D. E. Fragment Hits: What do They

Look Like and How do They Bind? *J. Med. Chem.* **62**, 3381–3394, 2019]. Based on more than 1200 protein-fragment complexes we estimated the maximal available surface is around 20 Å² per heavy atom. Taking into account that the estimated free energy gain of 20 Å² corresponds to 1 kJ/mol [Olsson, T. S. G., Williams, M. A., Pitt, W. R. & Ladbury, J. E. The Thermodynamics of Protein-Ligand Interaction and Solvation: Insights for Ligand Design. *J. Mol. Biol.* **384**, 1002–1017, 2008], we see that no more than 1 kJ/mol per heavy atom can be gained by desolvation. As rigid-body entropy loss upon binding is estimated to be 15–20 kJ/mol [Murray, C. W. & Verdonk, M. L. The consequences of translational and rotational entropy lost by small molecules on binding to proteins. *J. Comput. Aided. Mol. Des.* **16**, 741–753, 2002], this suggests that in most cases apolar desolvation alone is unable to ensure the effective binding of fragments. Consequently, specific polar interactions are mainly responsible to ensure higher affinity binding and, practically, the observation of the binding event. Nonetheless, apolar desolvation effects are implicitly considered in pharmacophores that contain features of the type *H* (hydrophobic group) or *R* (aromatic ring): particularly for the former, the contribution to ligand binding includes apolar desolvation.

The next aspect of selecting the features is their relative contribution to the binding free energy of the fragment. Therefore, we discuss the role of the interaction energy based ePharmacophore approach in selecting the final set of features encoded in the pharmacophore pattern in the Supplementary Information, section 1.2.

6. The SpotXplorer0 library has been tested on five known cases and the fragment selection has been performed based on a biochemical assay. Fragments are often difficult to detect due to their weak binding. Therefore, what was the sensitivity level of the applied assays? Some information has to be provided. For the two proteases a fluorogenic substrate, which gives a more sensitive assay, might have provided a larger number of hits.

Considering that the set of investigated GPCRs has fragment like endogenous ligands, fragments provide reasonable affinity and therefore assay results should not be influenced significantly by the assay sensitivity. For fragment screening against GPCR targets, competitive binding assays on cell lines with overexpression of the individual protein, and specific radioligands were used, providing high sensitivity [Visegrády, A. & Keserű, G. M. Fragment-based lead discovery on G-protein-coupled receptors. *Expert Opin. Drug Discov.* **8**, 811–820, 2013]. Radioligand displacement even by a high concentration of a low-affinity fragment is easily measured in these settings.

The protease assays used here are kinetic enzymatic assays, measuring the initial velocity of the enzymatic reaction and not only a single-point value at a given time. Thus, they are robust and generally offer good reproducibility, and they can also be used for screening compounds at high concentrations. While they might have a limitation of sensitivity, in our specific setup (substrate concentration) we were able to reliably (and reproducibly) measure compounds even with millimolar activities (which was also used as a criteria for considering compounds as active). Using a more sensitive fluorogenic assay would generate more hits, but these would be weaker binders and therefore of limited interest (and out of criteria used for this manuscript).

See also our answer to remark 5 of the second reviewer.

7. The part on the comparison with the binding pharmacophores present in ChEMBL is

hardly comprehensible with the given information. The authors are requested to reformulate this part and to provide more insights in the analysis.

Thank you for pointing this out, we have added a more detailed explanation for the Results section, subsection “*SpotXplorer screening yields selective fragment hits for GPCRs and proteases*”, paragraphs 2-4.

8. The authors mention that aromatic fragments are preferentially selected. The question is whether this observation reflects a “real” preference or results from the above-mentioned “self-consistency”. Fragments with aromatic groups have been selected in the past due to their easy chemical access. Is this aspect finally reproduced due to a biased pharmacophore preselection? At least the reader can expect some critical discussion by the authors.

Accepting the comment of the referee, we mention here the published analyses of fragment hits [Ferency, G. G. & Keserú, G. M. Thermodynamics of Fragment Binding. *J. Chem. Inf. Model.* **52**, 1039–1045, 2012; Giordanetto, F., Jin, C., Willmore, L., Feher, M. & Shaw, D. E. Fragment Hits: What do They Look Like and How do They Bind? *J. Med. Chem.* **62**, 3381–3394, 2019] and also the results discussed by Chris Swain (https://www.cresset-group.com/wp-content/uploads/2015/06/Fragment-based-screening-what-can-we-learn-from-published-hits_Chris-Swain.pdf), pointing out that a vast majority of fragment hits contain aromatic rings. In addition to the rigidity and hydrophobic character of these scaffolds, synthetic accessibility might also validate their frequent appearance in fragment hit lists (and also in published protein-fragment complexes). However, these analyses did not consider the impact of the aromatic moieties on fragment binding.

Also, our wording was likely misleading: the cited sentence was supposed to state that from the SpotXplorer0 library, partially aromatic fragments (containing both aromatic and non-aromatic rings) were preferentially identified as experimental hits (as represented by the light green color in the ring plots) i.e. we did not deliberately select them. This was clarified in the manuscript.

Realizing that by “selection”, the reviewer probably meant the selection of the compounds to constitute the SpotXplorer0 library, we note that about 25% of the total library consists of fully aromatic compounds (light blue ring portion in Figure 3A), with most of the compounds containing both aromatic and non-aromatic rings (light green ring portion in Figure 3A). This is in accordance with the fact that 47% of the 2- and 3-pt. pharmacophores (which were used for selecting the SpotXplorer0 library) do not contain an aromatic ring as a pharmacophoric feature (a related analysis was added to the Supplementary Information as section 1.6).

9. The analysis of the GPCR and protease data remains not really conclusive. How novel are the obtained fragments poses? Without a crystal structure this essential question remains unanswered. The best library suggests a number of fragments to the user which indicate deviating binding poses in a protein environment with different orientations of exit vectors for synthesis. Also fragments with optimal pharmacophore distributions do not help much further if overlapping binding modes reproducing the same interaction patterns are produced. It is highly suggested to the authors to add another crystallographic screen for at least one or two of the targets. Particularly the two proteases Factor Xa and Thrombin are well established and could be easily screened with the facilities present at Diamond within

very short time. This extension of the current study is highly suggested to the authors and the journal's editor.

Analysis of the ChEMBL data for GPCRs and proteases was aimed at the collection of known pharmacophores. We asked what percentage of the known pharmacophores was covered by the pharmacophores identified by screening the optimized SpotXplorer0 library. This analysis revealed that 50-80% of the known pharmacophores was explored. Here our goal was to demonstrate the effectiveness of pharmacophore mapping at binding hot spots that might be indicative to other targets. This was further clarified in the main text, also in accordance with comment 7 of this reviewer. Since there was no structural information available for either ChEMBL ligands or SpotXplorer fragments with the actual targets, binding poses could not be analyzed and therefore discovery of new pharmacophores could not be considered.

While we agree that X-ray screening on further targets would be a valuable addition to the manuscript, upon consulting with the relevant XChem co-authors and infrastructure management, we currently do not see a way to complete this point as suggested, due to the following practical reasons:

1. Crystallographic fragment screens were performed using the unique infrastructure of the Diamond Light Source, UK. This facility is made available to researchers through a competitive application process and shared across multiple teams based on strict schedules. Hence, the beamlines are at the disposal of the XChem team for a limited amount of time, which is to be distributed according to pre-defined criteria of importance, relevance and novelty.
2. Most importantly, last (and also this) year, due to the COVID-19 pandemic, each of the scheduled fragment screens (mostly on relevant new oncotargets) were postponed to give preference to research on SARS-CoV-2-related targets. This is still in effect, with fragment libraries being screened against a growing selection of viral target proteins. Our support from the Foreign Commonwealth and Development Office, UK and National Office of Research, Development and Innovation, Hungary are dedicated to these COVID-19 related efforts.
3. Even if there was no pandemic, it would be difficult to justify the allocation of limited beamline capacity to such well-characterized protein targets as thrombin and Factor Xa. With a large number of crystal structures being available for both proteins in the PDB database (more than 100 for Factor Xa and more than 1000 for thrombin), additional hit discovery efforts against these targets would be unlikely to result in sufficient novelty, and the chance for medicinal chemical follow-up is likewise very low.
4. Being the only public, high-performance research facility dedicated for crystallographic fragment screening in the world, we cannot envision to complete this suggestion by involving other groups, most importantly due to the lack of the high-power synchrotron facility of Diamond which is necessary to detect fragment-sized (and therefore relatively weak) binders.
5. It is worth to note that despite of (or, in line with) these limitations, we performed full crystallographic screens against two highly relevant and challenging targets including the 3CL main protease, a key viral protein picked up by many drug discovery programs recently, and NSP3, the essential component of the replication and transcription complex of SARS-COV-2 that might serve as a novel target for antiviral drugs.

In the meantime, we realize the importance of this reviewer feedback and we are committed to do every reasonable effort to answer the question raised here. Therefore, we initiated an editorial consultancy with the responsible Editor of Nature Communications, who agreed that the additional X-ray screen would go beyond scope. We suggested an alternative approach

that was tentatively accepted and which will hopefully also meet the expectations of this reviewer:

1. The point of the reviewer's suggestion is to verify that the fragment hits represent different binding modes, and are thus non-redundant. Our methodology is focusing on the key binding hot spots since these are the most conserved regions that can be described by a limited set of pharmacophores. Therefore, our library was optimized for covering these pharmacophores and the main goal of screening this library is identifying fragments bound to the hot spots of the target. Although different binding modes were not an explicit goal in this context, this is actually already verified by the NSP3 fragment hits (as the same reviewer mentions in his next, 10th, remark). Here, we included a new figure (Figure 5) to highlight the point raised by the referee, and added more detail to the relevant part of the discussion, which now mentions the relevant pharmacophores and possible growing vectors.

2. For Factor Xa and thrombin, even though we do not have crystal structures, the possible poses of the fragment hits are determined by the available arrangement of their pharmacophore features since these are the structural moieties forming the critical interactions with the surrounding binding site residues. Therefore, we performed a comprehensive and comparative analysis of X-ray validated binding pharmacophores extracted from all available thrombin and Factor Xa structures and those pharmacophores identified by screening the SpotXplorer0 library against these targets.

The results of this analysis were now added to the manuscript (Results, "*SpotXplorer screening yields selective fragment hits for GPCRs and proteases*" subsection, paragraphs 6-9).

10. The 3CLPro and NSP3 study underlines the importance of performing a crystallographic validation of the ScoreXplorer0 library and may disclose its advantage over other libraries as it seems that the found entries suggest alternative binding features even though they bind in an overlapping region of the binding site.

Our library was designed to reproduce the binding pharmacophores identified at protein hot spots. Therefore, when assembling the non-redundant set of binding pharmacophores, we first checked whether the co-crystallized fragment is located at the binding hot spot predicted by FTMap. Since we picked up only those pharmacophores identified at the mostly conserved binding hot spots, the objective of protein mapping by this library is identifying the key hot spot.

We checked the binding site of the non-covalent fragments found in the respective fragment screens at Diamond for 3CLPro [Douangamath et al. Crystallographic and electrophilic fragment screening of the SARS-CoV-2 main protease. *Nat. Commun.* **11**, 5047, 2020] and NSP3, and compared to those identified from SpotXplorer0. This comparison revealed that screening the SpotXplorer0 library identified the key hot spot (active site) of these proteins, typically used by the endogenous ligands. In the figures below (taken from the Diamond Light Source website, <https://www.diamond.ac.uk/covid-19/for-scientists/NSP3-macrodome-structure-and-XChem/XChem-Fragment-Screen.html>), these are the most populated sites (labeled as "adenine site" for NSP3).

3CLPro

NSP3

In addition, screening against NSP3 identified alternative binding modes at this site. We have highlighted this in the new Figure 5 and the relevant part of the discussion.

11. It is surprising the authors do not provide, as in the other cases, quantitative assay results for the ligands of the NSP3 macrodomain and only describe that they “show values in the high micromolar range”. The missing information should be provided.

The “high micromolar range” refers to the cellular assay and these results are actually included in Figure 4 (for SX051), Supplementary Information section 10 (SX005, SX048), but not for SX003 and SX054 which were inactive in cells.

Accepting the comments of the referee, we also tried to evaluate the NSP3 inhibitory activity of the co-crystallized fragments in an HTRF assay. NSP3 inhibition was assessed by the displacement of an ADP-ribose-conjugated biotin peptide from the His₆-tagged macrodomain using HTRF with a Eu³⁺-conjugated anti-His₆ antibody donor and streptavidin-conjugated acceptor. Technical details of the protocol are now included in the Supplementary Information, section 9.

These measurements revealed that SX005 has an IC₅₀ of 465 μM, SX048 showed significant inhibition at 5 mM concentration, while SX003, SX051 and SX054 showed no activity in the enzyme assay up to 5 mM concentration. These data are now added to the Supplementary information (section 9).

12. The chemical formulas of the entries of the fragment library have to be documented at least in the Supporting Information. This is definitely a prerequisite for acceptance of this contribution as only then it is in accordance with good scientific practice.

Structures were originally included as SMILES codes (Supplementary Data excel file, “SpotXplorer 0” sheet), but were now added as images, too (Supplementary Data, “SpotXplorer 0 – misc.” sheet).

13. The citations are somewhat biased to the publishing authors. Unfortunately, the Nature series does not allow for the full author list. In the current case, the authors sometimes provide a full author’s list, sometimes they shortcut to the first author, which usually is the one who did the work but is the unknown PhD student of a group. Without retrieving the full information from a library database, it remains unclear to the reader from which group a

particular contribution originated. This is a rather unfortunate service to the scientific community by the journal.

The reference list is updated to the Nature (no “et al.”) format, which lists all of the authors.

With the above-described revisions the paper should be reconsidered for publication in Nature Communication.

Again, we are grateful to this reviewer for the suggestions that helped us improving our manuscript further.

Reviewer #2 (Remarks to the Author):

It has long been known that some endogenous and exogenous molecules target several proteins. Drug discovery has taken advantage of this phenomenon through the development of privileged scaffolds, or the repositioning of drugs. Analysis of the 3D-structures in the Protein Data Bank suggested physical basis of molecular promiscuity. Many fragments have versatile binding properties, as do some druglike ligands. In addition, a limited number of protein pockets can cover the pocketome.

Bajusz et al. skillfully exploited the knowledge about protein/fragment recognition and issued a chemical library which is, to my knowledge, completely original.

The three major assets of this chemical library are:

- Its small size (96 fragments): evaluation of the biological activity of compounds is possible by low throughput biophysical methods
- Its universality: it is suitable for all kinds of target protein
- Its annotation: the pharmacophore models covered by each fragment are indicated

The manuscript first introduces the library design, then demonstrates its usefulness in screening exercises. The manuscript is well written. The message delivered is clear and the results remarkable. The authors have also provided the exact constitution of the chemical library. It is therefore possible to purchase the fragments from suppliers.

Thank you for the positive evaluation of our work and for the constructive suggestions.

****About the method for designing the library**** The authors used proven methods, intelligently combined. I only have minor questions and suggestions:

1. Were there requirements about a minimal number of polar / directional probes in pharmacophore models?

This question is very much on point, since directional (H-bond) features (and also ionic centers) play a crucial role in driving the thermodynamics of fragment binding, as evidenced in our earlier works. However, we did not set such criteria, in order to represent all possible, experimentally confirmed binding pharmacophore arrangements, with the minimum of 2 and the maximum of 4 features. To reflect on the relevant comment of the referee, here we analyzed what percentage of pharmacophores contain certain features. The result of the analysis is now included in the Supplementary information, section 1.6. Notably, 42% of 2-pt. pharmacophores does not contain any directional (H-bond) features: this drops to 24% and 5% for 3-pt. and 4-pt. pharmacophores, respectively (12% for the total set of pharmacophores). If we also account for other polar features (negative or positive ionic centers), then there are only 18/6/1% of 2/3/4-pt. pharmacophores that do not contain any of these (3% for the total set of pharmacophores). This is in line with our previous observation that apolar desolvation of fragments alone cannot compensate for rigid-body entropy loss upon binding (15-20 kJ/mol, vs. maximum 1 kJ/mol per heavy atom gained by desolvation) [Ferenczy, G. G. & Keserű, G. M. Thermodynamics of Fragment Binding. *J. Chem. Inf. Model.* **52**, 1039–1045, 2012]. By contrast, 41% of the pharmacophores do not contain any apolar

features (hydrophobic moieties or aromatic rings). These observations point to the conclusion that in order to ensure higher affinity binding, the hot spot must provide specific polar interactions, resulting in the binding of fragments to be typically enthalpy-driven [Ferenczy, G. G. & Keserű, G. M. Thermodynamics of Fragment Binding. *J. Chem. Inf. Model.* **52**, 1039–1045, 2012].

2. About FTMap, did the authors consider all predicted hotspots, or only the hotspots in clusters? How FTMap results were used?

Here, we performed FTMap analysis on the protein-fragment complexes and considered only fragments bound to the three highest ranked hotspots. For each of the fragment-bound structures, we mapped an individual chain with FTMap. We then compared the location of the top three FTMap hotspots to the location of the fragment. If the fragment was located within 3 Å of a top ranked hotspot, we considered the fragment to be bound to a hotspot in the protein. This approach reflects on evolutionary selection, since there is a limited number of endogenous ligands, targeting a limited number of binding sites. This was updated in the Supplementary information (section 1.3).

3. Pharmacophore models are in 3D. What is the maximal number of conformers per fragment? How many conformers of a fragment have to match a pharmacophore model to conclude that the fragment matches this pharmacophore model?

For screening the fragments against the pharmacophore models, we have conducted a full conformational search with Macromodel, with a mixed torsional/low-mode sampling protocol (with enhanced torsional sampling), max. 200 steps (max. 20 per rotatable bond), an energy window of 15 kJ/mol and an RMSD cutoff of 1.0 Å (other settings were kept as their default values). Consequently, there was no fixed number of conformations: for each fragment, all conformations were kept that were no more than 15 kJ/mol above the “global minimum” (the lowest-energy conformation that was actually identified by Macromodel) and deviated by more than 1.0 Å in RMSD from all previously identified conformations. Since all of these are (relatively) low-energy conformers, one matching conformer was considered to be enough during screening.

This was updated in the Supplementary information (section 2.1) to add a bit more detail along this reviewer comment.

4. Reference 24: Dr. Erlanson's blog is an excellent source of information although unconventional (not peer reviewed). In the case it is not kept online, I suggest to name the 5 chemical vendors in the article.

The result of the poll depends on the feedback provided by the participants. In this case, they are less than 100 and they ranked the vendors by popularity. One can retrieve the 5 most popular vendors from the blog. We do provide coverage information for these companies; however, we would not like to link this to particular vendors due to business sensitivity. For the sake of completeness, we disclose this information here, but definitely not in the paper.

A – Maybridge, B – Chembridge, C – Enamine, D – LifeChem, E – KeyOrganics (see supp. section 4)

We have added a brief comment („vendor names undisclosed due to business sensitivity”) to the respective figures in Supplementary Information section 4).

5. ****About the screening hits**** The library was designed to cover the maximum number of “true” pharmacophore models with a minimum number of fragments. Therefore, it is suitable for identifying hits at any protein site. The proof of concept was made on eight proteins of different class and biological functions. Five of them are widely studied targets, and therefore could be viewed as “easy”. Noteworthy, the authors also worked on three original targets (including two SARS-COV-2 proteins). Active fragments were obtained for all the proteins. The results are very impressive, given the small size of the chemical library. Different experimental assays were used (cell-based binding, activity of recombinant enzyme, Xray crystallography). Biochemical and functional assays have a high propensity to generate false positive, especially when testing compounds at high concentration (here from 10 micromolar to >500 micromolar). Was this point addressed by a confirmatory assay? Could the authors comment on the validation of hits?

To address this comment as fully as possible, we first summarize common sources of assay interference considered in this study, then reflect additionally on the specific bioassays contained in the manuscript. Sources of general assay interference include:

- a) Precipitation of the compounds, which can be detected visually (and it did not occur in the case of the SpotXplorer compounds since screening concentrations were always below the solubility limit of the fragments). Notably, this can lower the sensitivity of UV/Vis readouts (as the initial absorbance is higher).
- b) Compounds with “PAINS” activity, which would inhibit the enzyme in a non-specific fashion [Baell, J. B. & Holloway, G. A. New substructure filters for removal of pan assay interference compounds (PAINS) from screening libraries and for their exclusion in bioassays. *J. Med. Chem.* **53**, 2719–2740, 2010]. This is unlikely for SpotXplorer compounds since PAINS filtering was performed prior to assembling the library [Saubern, S., Guha, R. & Baell, J. B. KNIME Workflow to Assess PAINS Filters in SMARTS Format. Comparison of RDKit and Indigo Cheminformatics Libraries. *Mol. Inform.* **30**, 847–850, 2011].
- c) Aggregates: several nonspecific compounds form submicrometer aggregates that can inhibit many different enzymes, *via* direct association of the proteins with the surface of such aggregates [McGovern, S. L., Helfand, B. T., Feng, B. & Shoichet, B. K. A specific mechanism of nonspecific inhibition. *J. Med. Chem.* **46**, 4265–4272, 2003]. Aggregation-related non-specific inhibition of these proteins is unlikely:
 - i) We used the same assay conditions regarding compound concentration, buffer, *etc.* for both proteases, and almost the same conditions for the three GPCRs (with minor differences in buffer composition), thus we can regard each assay also as a control for the other assay(s) of the group. If several hits would be doubled (*e.g.* would possess similar activity on more targets), that would implicate non-selective inhibition based on some property of the compounds (*e.g.* aggregates). This was not the case here, with the exception of the four fragment hits of the 5-HT₇ receptor, which is known to have similar

pharmacophores to 5-HT₆ [Glennon, R. A. Higher-end serotonin receptors: 5-HT₅, 5-HT₆, and 5-HT₇. *J. Med. Chem.* **46**, 2795–2812, 2003].

ii) Aggregators are typically larger molecules with clogP values over 3 [Irwin, J. J., Duan, D., Torosyan, H., Doak, A. K., Ziebart, K. T., Sterling, T., Tumanian, G. & Shoichet, B. K. An Aggregation Advisor for Ligand Discovery. *J. Med. Chem.* **58**, 7076–7087, 2015]. Checking our fragment hits at <http://advisor.bkslab.org> (a web service for identifying likely aggregators) did not reveal any of them to be reported aggregators.

In addition, here we provide assay-specific details on hit validation. In radioligand binding assays (GPCR targets), the tested fragments directly compete with the isotope-labelled reference compound (in the system overexpressing the biological target) for binding to a given receptor binding site, therefore these kinds of assays are generally free of false positive cases. Additionally, we have used as a positive control a highly active ligand in high (saturation) concentration (10 μM of 5-HT in 5-HT_{1A}R and 5-HT₇R binding experiments, 10 μM of mianserin in 5-HT₆R assays); and the vehicle without a tested compound as a negative control (membrane homogenate, radioligand, assay buffer, DMSO).

The protease inhibitory assays (Thrombin, Factor Xa) are kinetic enzymatic assays, measuring the initial velocity of the enzymatic reaction (not only single-point response values at a given time). Thus, they are robust and generally offer good reproducibility, and can be used also for screening compounds at high concentration. Here, a specific source of interference would be compounds absorbing at the same wavelength as the cleaved chromophore ($\lambda = 405 \text{ nm}$ / yellow color of the solution), which lowers the sensitivity of direct readout, as the initial absorbance is higher. However, this does not generate false positives as we measure the slope of the initial, linear part of the curve (initial velocity of enzymatic reaction). Positive and negative controls were applied here as well, containing a marketed drug (Thrombin – dabigatran, FactorXa – rivaroxaban) and vehicle, respectively.

The SETD2 activity chemiluminescent assay kit is designed to measure SETD2 activity by quantification of the levels of H3K36me3 via antibody binding. As SETD2 is the only enzyme triggering H3K36 tri-methylation, this assay system is very specific and well-suited to identify agents inhibiting the enzymatic activity of SETD2. For the validation of on-target activity in the SETD2 chemiluminescence assay, we used Sinefungin, a previously reported inhibitor of the SETD2 histone methyltransferase (Zheng et al., *J. Am. Chem Soc.* 2012). Addition of Sinefungin resulted in complete inhibition of SETD2 activity in this assay. Therefore, the degree of SETD2 inhibition of selected compounds was calculated by normalizing the values to a negative control (no inhibition of SETD2 activity) and a positive control (200 μM Sinefungin, full SETD2 inhibition). The cell-based CellTiter-Glo assay benefits from its robust, ready-to-use reagent, as multiple pipette steps are not required. The generation of a luminescent signal is directly proportional to the amount of ATP present in the sample, hence also to the number of cells in the culture well. For the cell-based validation of fragment hits in leukemia cells, selected compounds were tested for their antiproliferative activity in cell lines that were previously shown to be sensitive to perturbation of SETD2 (Skucha et al., *Nat. Commun.* 2018). To avoid DMSO-induced toxicity that is observed at concentrations higher than 0.1%, compounds were diluted from 500 mM stock solutions.

The *in vitro* SARS-CoV-2 cell-based antiviral assay utilized virus-specific detection methods. Only the viral RdRp gene can be detected with the used Droplet-Digital PCR system. This method has been validated previously by the members of the National Laboratory of Virology,

Pécs. Furthermore, prior to the Droplet- Digital PCR viral copy number quantification, the Vero E6 cells were observed whether there are visible virus-induced cytopathic effects on them. For measuring SARS-CoV-2 3CLPro inhibition, the RapidFire SPE-MS assay was used, which is robust, with a low signal and $Z' > 0.8$. With RapidFire being a mass-based measurement, one of the biggest advantage is reduced false positives as opposed to fluorescence-based assays, since specific masses are detected.

Minor: In July, the WHO announced that it has discontinued trials of the HIV drugs as a treatment for COVID19 patients.

We have updated this as follows: “Viral protease inhibition is a well-established strategy for the treatment of viral diseases such as HIV, and HIV protease inhibitors have entered into early clinical trials against SARS-CoV-2³⁷ as well as MERS-CoV³⁸ (although the lopinavir/ritonavir trial against COVID-19 was discontinued in July 2020 due to ineffectiveness).³⁹” (With an added reference to the interim report of the WHO Solidarity Trial Consortium: 10.1056/nejmoa2023184)

Last, could the authors comment on the solubility of the 96 fragments in water?

Typically SpotXplorer compounds were dissolved in DMSO to provide 200 mM DMSO stock solutions and were diluted 1:100 into assay buffers. This is closest to a kinetic solubility assay that confirmed the minimal solubility of 2 mM. In addition, we compiled all the relevant information that is available to us regarding solubility, into the Supplementary Data file (“SpotXplorer 0 – misc.” sheet). This includes:

- calculated logS (aqueous solubility) values (Schrödinger QikProp)
- calculated logD (octanol-water partition coefficient, accounting for ionization) values at a pH of 7.4 (Chemaxon)
- highest assay concentration: the highest concentration in which the compound was assayed (either in this work, or in other projects if we know of such). The compounds were dissolved primarily in DMSO and then diluted into an aqueous buffer (DMSO content did not exceed 1% in most of the assay conditions).

Reviewer #3 (Remarks to the Author):

This is a very important manuscript for the drug discovery community for which a substantial part of the novel chemical matter identified across all targets originates from fragment approaches. The manuscript describes an elegant selection and construction of a library of fragments that represents binders of common pharmacophores in proteins. The selection of this 96-member fragment library enabled the successful identification of fragments against two established protein targets (GPCR, protease) as well as with three, more difficult targets (human SETD2, 3CLPro and NSP3 from Sars-CoV2). Especially the identification of initial chemical matter binding to the latter two targets is a very valuable confirmation of the performance of this new fragment library selection.

This paper - and the new fragment library - will be of tremendous use to the fragment-based drug discovery community as it allows with high reliability the identification of fragment hits from a very small collection of just 96 molecules.

Thank you for the positive evaluation of our work.

Reviewers' Comments:

Reviewer #2:

Remarks to the Author:

You have answered all my questions in great detail and very convincingly. Thank you.